# Physical Literacy, Physical Activity, and Health Indicators in School-Age Children

**DOI:** 10.3390/ijerph17155367

**Published:** 2020-07-25

**Authors:** Hilary A.T. Caldwell, Natascja A. Di Cristofaro, John Cairney, Steven R. Bray, Maureen J. MacDonald, Brian W. Timmons

**Affiliations:** 1Child Health and Exercise Medicine Program, Department of Pediatrics, McMaster University, Hamilton, ON L8S 4L8, Canada; caldweha@mcmaster.ca (H.A.T.C.); dalimona@mcmaster.ca (N.A.D.C.); 2Department of Kinesiology, McMaster University, Hamilton, ON L8S 4L8, Canada; sbray@mcmaster.ca (S.R.B.); macdonmj@mcmaster.ca (M.J.M.); 3School of Human Health and Nutritional Sciences, University of Queensland, St Lucia QLD 4072, Australia; j.cairney@uq.edu.au

**Keywords:** youth, aerobic fitness–body composition, blood pressure, quality of life, mediation

## Abstract

It has been theorized that physical literacy is associated with physical activity and health. The purpose of this study is to investigate the associations between physical literacy and health, and if this relationship is mediated by moderate-to-vigorous physical activity (MVPA). Two hundred and twenty-two children (113 girls, 10.7 ± 1.0 years old) participated in this cross-sectional study. A physical literacy composite score was computed from measures of PLAYfun, PLAYparent, and PLAYself. Physical activity was measured over seven days with accelerometers, expressed as MVPA (min/day). Health indicators included: body composition (percent body fat), aerobic fitness (treadmill time and 60s heart rate recovery), resting systolic blood pressure, and quality of life. Physical literacy was significantly associated (*p* < 0.001) with percent body fat (R^2^ = 0.23), treadmill time (R^2^ = 0.21), 60 s heart rate recovery (R^2^ = 0.36), systolic blood pressure (R^2^ = 0.11), and quality of life (R^2^ = 0.11). The relationships between physical literacy and aerobic fitness, but not other health indicators, were directly mediated by MVPA. Higher physical literacy in children is associated with favorable health indicators, and the relationships between physical literacy and aerobic fitness were influenced by MVPA. Future work should examine these relationships longitudinally and determine if changes in physical literacy leads to changes in health.

## 1. Introduction

Physical literacy is theorized to be the foundation of lifetime physical activity participation and is defined as the motivation, confidence, physical competence, knowledge and, understanding to value and take responsibility for engagement in physical activities for life [1,2,3]. This definition of physical literacy includes four interconnected elements (affective, physical, cognitive, and behavioral) that change and adapt across the lifespan [3]. In childhood, increased physical activity participation is associated with numerous health benefits, including decreased adiposity, reductions in cardiometabolic disease risk, increases in aerobic fitness and muscular strength, and higher quality of life [4]. The majority of Canadian children and youth are not participating in enough physical activity to achieve health benefits and innovative strategies are needed to increase participation [5]. If physical literacy is the gateway to increasing physical activity, then physical literacy may be an indirect determinant of health, as increased physical activity is associated with health benefits [6]. The enthusiasm for physical literacy in physical education, public health, sport, and recreation has out-paced research on this topic [7]. As such, further empirical evidence about the relationships between physical literacy, physical activity, and health indicators is necessary to advance knowledge in this field. 

In Margaret Whitehead’s text, Physical Literacy Throughout the Lifecourse, she theorizes that physical literacy ought to be associated with weight status, fitness, physical activity, and motor competence [2]. More recently, it has been suggested that physical literacy be considered a determinant of health through the following reciprocal pathways: elevated physical literacy leads to greater physical activity participation, which leads to positive physiological, social, and psychosocial adaptations, resulting in improved physical, mental, and social health; this pathway would be present and dynamic across the lifespan from early childhood to old age [6]. Another important part of this proposed model is that some relationships are bidirectional. For example, physical literacy is proposed to be an important determinant of physical activity participation, but development of physical literacy is suggested to occur through structured and unstructured physical activity opportunities [6]. As this proposed pathway is novel, limited empirical research has explored these relationships.

Based on the above definitions and theories, physical literacy has gained attention as the foundation for lifelong physical activity participation, which would, in turn, lead to desirable physical and psychosocial benefits [3,6,8,9]. Several studies have outlined these associations in a large sample of Canadian children whose physical literacy was assessed with the Canadian Assessment of Physical Literacy (CAPL) [10,11]. In those studies, healthy weight children demonstrated slightly higher overall physical literacy and higher scores in modified physical competence, daily behavior, motivation and confidence, and knowledge and understanding domains than children who were overweight or obese [12]. Children and youth who met the Canadian Physical Activity Guidelines of 60 min of daily moderate-to-vigorous physical activity (MVPA) demonstrated higher physical competence, and motivation and confidence scores than those who did not meet the Physical Activity Guideline, while the knowledge and understanding score was not associated with meeting the Physical Activity Guideline [13]. Higher physical literacy scores were associated with higher cardiorespiratory fitness, but the associations between physical literacy and other aspects of fitness (i.e., muscle endurance and muscle strength) were not studied [14]. 

The Physical Literacy Assessment for Youth (PLAY) tools are another measure of individual physical literacy and includes several workbooks to assess different domains of physical literacy [15]. PLAYfun is a measure of individual movement competence with several domains, including running, locomotor, upper and lower body object control, and balance [15]. Physical activity participation (measured with pedometers) was significantly associated with the PLAYfun total score, locomotor domain score, and balance domain score in 7–14 year-old children [16]. The relationships between physical literacy, as assessed with the PLAY Tools, and fitness, body composition, blood pressure, and health-related quality of life have not been previously studied. 

In light of the gaps in knowledge, the purpose of this study was to examine the cross-sectional associations between physical literacy and body composition, fitness, blood pressure, and health-related quality of life, and to determine if these relationships are mediated by physical activity participation in Canadian school-age children (8–13-year-old children). It was hypothesized that physical literacy would be positively associated with aerobic fitness and health-related quality of life, negatively associated with body composition and blood pressure, and that these relationships would be mediated by physical activity participation. 

## 2. Materials and Methods

### 2.1. Participants and Design

Participants in this study were part of the school-age kids health from early investment in physical activity (SKIP) study, a 3-year longitudinal cohort study of physical activity and health outcomes in school-age boys and girls. The SKIP study was a follow-up to the health outcomes and physical activity in preschoolers (HOPP) study, described previously [17]. At enrollment, children with diagnosed medical conditions or known developmental or cognitive delays were excluded. Physical literacy assessments were added to the third and final year of the SKIP study. The Hamilton Integrated Research Ethics Board provided ethical approval for the study. All parents provided informed, written consent and all children provided written, informed assent. 

### 2.2. Measures

#### 2.2.1. Years from Peak Height Velocity (YPHV)

YPHV was calculated with validated equations that included the following variables: gender, date of birth, date of measurement, standing height, sitting height, and weight. Assessment of YPHV is a non-invasive, practical method to assess maturity status and the equations have been tested and cross-validated in longitudinal samples. The mean difference between actual and predicted maturity was 0.243 ± 0.650 years for boys and 0.001 ± 0.678 years in girls, allowing an accurate prediction of biological age [18]. Due to the age of participants in this study, models were adjusted for YPHV, rather than for chronological age. 

#### 2.2.2. Body Mass Index (BMI)

Height and weight were measured using standard procedures [17]. BMI was calculated as weight/height^2^ (kg/m^2^). BMI percentiles, based on sex and age, were calculated using Centre for Disease Control growth charts for descriptive purposes [19].

#### 2.2.3. Physical Literacy

The PLAY Tools were developed by Sport for Life and represent a series of assessment tools to assess the multiple domains of physical literacy [15]. The PLAY Tools were designed for children 7 years and older. In combination, the PLAYfun, the PLAYself, and the PLAYparent tools provide a multi-perspective assessment of a participant’s physical literacy [15]. Participants in the SKIP Study completed PLAYfun and PLAYself and a parent or guardian of each participant completed PLAYparent. 

The PLAYfun assessment includes 18 movement skills within five domains: running, locomotor, object control (upper body, object control) lower body, and balance, stability, and body control. PLAYfun was administered with the same methods as previously described [7,20]. The total score is the average score of all 18 task scores [15]. All PLAYfun assessments were administered and scored by one of two investigators (HC and ND). 

The PLAYself questionnaire is a 22-item self-evaluation of a child’s perception of their own physical literacy [15]. The PLAYself questionnaire includes four subsections: environment, physical literacy self-description, relative rankings of literacies (literacy, numeracy, physical literacy) and fitness. The PLAYself score was calculated by adding up the totals of the subsections and dividing by 27, as outlined in the PLAYself workbook [15].

The PLAYparent questionnaire was used to assess a parent’s perception of their child’s level of physical literacy, including questions about the child’s ability, confidence, and participation. PLAYparent provided researchers with an additional perspective and identified positive and negative factors that affect the child’s ability to lead a healthy lifestyle. The PLAYparent questionnaire is divided into five subsections: physical literacy VAS, cognitive domain, environment, motor competence (locomotor and object control) and fitness [15]. The PLAYparent questionnaire was scored by summing the parents’ responses and multiplying by 2.63 to give a total out of 150, as outlined in the PLAYparent workbook [15]. 

A physical literacy composite score was calculated using the standardized scores of PLAYfun, PLAYparent, and PLAYself. The standardized scores were summed, with higher values suggesting greater physical literacy. 

#### 2.2.4. Body Composition

Percent body fat (%BF) was measured by bioelectrical impedance analysis (BIA; RJL Quantum 2, Tanita Corporation, Japan). Fat free mass (FFM) was calculated using an equation that was validated against DEXA in children [21]; %BF was then calculated as ((body weight- FFM)/body weight] × 100). 

#### 2.2.5. Physical Activity

Physical activity was assessed using Actigraph GT3X accelerometers (Fort Walton Beach, FL, USA). The accelerometers recorded raw accelerations at 30 Hz during waking hours for seven days, except during swimming or bathing. Participants wore the accelerometer on a belt over their right hip. Participants and/or parents were instructed to record the times the accelerometer was put on and taken off in the provided logbook. Accelerometer data were downloaded in 3s epochs, visually inspected for any spurious activity counts, and processed with Actilife Software (Version 6.11.9, Actigraph, Pensacola, FL). A non-wear period was defined as 60 min or more of continuous zero counts or if the logbook indicated device removal. Only participants who wore the accelerometer for at least 3 days with a minimum wear time of 10 h per day were included in the analyses. This minimum wear time provides a reliability coefficient of 0.9 for children [22]. Daily minutes of MVPA were calculated using Evenson et al. (2008) cut-points (≥574 counts/15-sec) [23] that are recommended to estimate time spent in different intensities of physical activity in children and adolescents [24]. Cut-points were divided by 5 to account for the 3 s epoch used in the current study [25]. 

#### 2.2.6. Aerobic Fitness

Aerobic fitness was assessed using a modified Bruce Protocol, a progressive treadmill test that increases in speed and grade every 3 min [26]. To ensure participant safety, participants were given the option to hold the handrails during the test and a researcher was positioned behind the treadmill. Participants were fitted with a heart rate (HR) monitor (Polar Electro, Kepele, Finland) to continuously monitor HR during the test and seated recovery. The test was terminated when the participants were exhausted, could no longer keep up with the speed of the treadmill and/or showed signs of emotional distress and/or refused to continue. In our sample, the average peak HR was 202 ± 7 bpm (183–226 bpm), suggesting participants were at, or near, exhaustion. Time to exhaustion with the Bruce Protocol is highly reproducible in school-age children (correlation coefficient = 0.94) [27]. Upon termination of the treadmill test, the participants were immediately seated and asked to remain as still as possible for 2 min. The second indicator of aerobic fitness was 60 s HR recovery (HRR), calculated as the difference between the peak HR (single beat highest value) and HR 60s into recovery. Higher values indicate faster autonomic recovery [28]. 

#### 2.2.7. Blood Pressure

Automated measures of seated blood pressure (Dinamap Pro 100; Critikon Inc) were obtained from the right arm at least 4 times with a 1 min delay between each measure. The 2nd, 3rd, and 4th measures were averaged if within 5 mmHg; additional measures were taken if the measures differed by more than 5 mm Hg [29]. Blood pressure was expressed as seated systolic blood pressure (SBP), as SBP is a better predictor of hypertension and cardiovascular events compared to diastolic blood pressure [30,31]. 

#### 2.2.8. Health-Related Quality of Life (HRQOL)

Participants completed the self-reported Pediatric Quality of Life (PedsQL) 4.0 Child Self-Report for 8–12-year-old children, a reliable, valid, 23-item questionnaire that evaluates children’s quality of life in 4 core domains: physical (8 items), emotional (5 items), social (5 items), and school functioning (5 items). Children were asked to rate how problematic each item had been in the previous month on a 5-point scale (never a problem, almost never a problem, sometimes a problem, often a problem, almost always a problem). The PedsQL outcomes are an aggregate score, with higher scores suggesting better HRQOL [32]. PedsQL cut-off scores for designating an at-risk status for impaired HRQOL and minimal clinically important difference values are also available to aid in the interpretation of results [33]. 

### 2.3. Statistical Analyses

All statistical analyses were performed in STATA (Version 14.2). A *p*-value of 0.05 was used to specific statistical significance. Descriptive statistics (mean, standard deviation, minimum and maximum) of the participant’s age, YPHV, sex, percent body fat, treadmill time, 60s HRR, blood pressure, HRQOL, and PLAY Tools were calculated. Sex-dependent variation in all measures were examined with *t*-tests. Physical literacy z-scores were calculated for PLAYfun, PLAYself, and PLAYparent as the individual values minus the group mean, divided by the standard deviation to achieve variables that had a mean of 0 and a standard deviation of 1. The physical literacy composite score was calculated as the sum of the PLAYfun, PLAYparent, and PLAYself z-scores. 

Linear regression was used to determine the relationships between physical literacy composite score and percent body, treadmill time, 60 s HRR, blood pressure, and HRQOL in independent models. Regression models were adjusted for participant’s sex and YPHV. Normality and skewness were assessed with the Shapiro–Wilk Test for Normality, the Skewness/Kurtosis Test for Normality, and visual inspection of P-P plots, Q-Q plots, and histograms. Collinearity between variables was assessed with the variance inflation factor. 

To further explore Cairney et al.’s model (2019), mediation analyses were conducted to determine if the associations between physical literacy and the various health indicators were mediated by MVPA. The tests for mediation effects were conducted independently for each health outcome using the SEM command in STATA. For each model, the physical literacy composite score was entered as the independent (X) variable, MVPA as the mediator (M) and health indicator as the dependent variable (Y), with sex and YPHV included as covariates. Bootstrapping was set to 10,000 samples [34]. Sex was included as a covariate because male children and youth engage in more habitual physical activity than females, as demonstrated by our results (Table 1) and in the literature [5]. YPHV was included because it is recommended that pediatric exercise science studies control for the effects of maturity on their results [35]. 

## 3. Results

### 3.1. Participants

Two hundred and forty-nine participants (121 girls, 128 boys) took part in the lab-based assessments of year 3 of the SKIP study, and 222 completed consent and assent forms to participate in the physical literacy assessments (113 girls, 109 boys, 10.7 ± 1.0 years). Only participants with consent and assent for the physical literacy assessments were included in subsequent analysis. Reasons for not participating in the physical literacy assessments included: the visit was scheduled before ethics approval was granted for these additional assessments (*n* = 3), participant and/or parent denied participation in the extra assessment (*n* = 19), or an assessor trained in the assessment of physical literacy was unavailable (*n* = 5). One participant who provided consent and assent became distressed in the visit and withdrew participation from PLAYfun and PLAYself, and the parents of two participants did not complete PLAYparent. Treadmill data (treadmill time and 60s HRR) from 11 participants was excluded from analyses for the following reasons: 1 participant did not participate in the treadmill test, 8 participants began the test but refused to continue and ended the test before reaching the termination criteria, and two participants stopped prematurely due to pre-existing musculoskeletal injuries. Two hundred and eight (93.7%) of the 222 participants met the accelerometer wear time criteria. Six children aged 8 did not complete the Peds-QL as per the study’s protocol that only participants ≥9 years old completed questionnaires. Two participants were missing blood pressure measures because one participant declined the measurement and one did not complete the vascular component of testing. Missing data were not imputed, and pairwise deletion was used for analysis. 

Descriptive statistics are included in Table 1. The Shapiro–Wilk test for normality showed that all variables, except 60s HRR, were not consistent with a normal distribution (*p* > 0.05). Visual interpretation of histograms further revealed that HRQOL (median: 78.26) and physical literacy composite (median: 0.07) were negatively skewed. Age was evenly distributed between 8–13 years old. YPHV, %BF, SBP, and MVPA appeared to be consistent with the normal distribution. Treadmill time was irregularly distributed and was not consistent with a normal or skewed distribution (median: 10.36). Variance inflation factors did not reveal collinearity among the independent variables. 

There were no differences between boys and girls in age, height, weight, BMI or BMI%ile (*p* = 294–0.904). Girls had smaller YPHV values, suggesting that they were more mature than boys (*p* < 0.001), and displayed a higher %BF than the boys (*p* < 0.001). The boys exhibited longer treadmill times (*p* = 0.005) and faster 60 s HRR (*p* < 0.001). The girls self-reported higher HRQOL total scores than the boys (*p* = 0.014), which were attributable to significantly higher psychosocial composite scale scores (includes emotional, social, and school functioning scales) in the girls versus the boys (76.58 ± 13.03 versus 72.02 ± 12.18, *p* = 0.008, respectively). On average, participants wore their accelerometers for 12.76 ± 0.70 h per day, and there were no differences in wear time between boys and girls. Boys participated in more MVPA than girls (*p* < 0.001). 

### 3.2. Physical Literacy and Health

Boys displayed higher PLAYfun scores than girls (Table 2; *p* = 0.017), but there were no differences in PLAYself and PLAYparent between boys and girls (*p* = 0.423–0.820). The physical literacy composite score ranged from −8.8 to 5.6, with no difference between boys and girls (*p* = 0.151). 

The physical literacy composite score was significantly associated with each health indicator (Table 3). The physical literacy composite score and YPHV were associated with %BF (R^2^ = 0.228, F (3,205) = 20.19, *p* < 0.001) and MVPA (R^2^ = 0.235, F (3,192) = 16.61, *p* < 0.001). The physical literacy composite score, sex, and YPHV were associated with SBP (R^2^ = 0.109, F (3,204) = 8.31, *p* < 0.001).

### 3.3. Mediation Analyses

As outlined above, physical literacy had a direct effect on all health indicators. Next, physical activity was explored as a mediator of the relationship between physical literacy and health indicators. With %BF as the dependent variable, there was a direct effect of physical literacy on MVPA (β = 3.20, 95% confidence interval (CI): 2.21–4.18, *p* < 0.001) and a non-significant direct effect of MVPA on %BF (β = −0.03, 95% CI: −0.07–0.001, *p* = 0.057). There was a non-significant indirect effect of MVPA on the relationship between physical literacy and %BF (β = −0.11, 95% CI: −0.22–0.003, *p* = 0.06), and this model explained 37% of the variance (R^2^ = 0.37) in %BF. Similar results were found for HRQOL, with a direct effect of physical literacy on MVPA (β = 3.17, 95% CI: 2.07–4.27, *p* < 0.001), a non-significant direct effect of MVPA on HRQOL (β = 0.03, 95% CI: −004, 0.10, *p* = 0.36), and a non-significant indirect effect of MVPA on the relationship between physical literacy and HRQOL (β = 0.10, 95% CI: −0.11–0.31, *p* = 0.36). This model explained 34% of the variance (R^2^ = 0.34) in HRQOL. For TM time and 60 s HRR, there was a direct effect of physical literacy on MVPA (TM Time: β = 3.04, 95% CI: 1.85–4.23; *p* < 0.001; 60 s HRR: β = 3.04, 95% CI: 1.85–4.23, *p* < 0.001), and a direct effect of MVPA on treadmill time (β = 0.04, 95% CI: 0.01–0.06, *p* = 0.002) and on 60 s HRR (β = 0.09, 95% CI: 0.02–0.16, *p* = 0.01). There was an indirect effect of MVPA on the relationship between physical literacy and treadmill time (β = 0.12, 95% CI: 0.02–0.22, *p* = 0.02), and between physical literacy and 60 s HRR (β = 0.27, 95% CI: 0.02–0.53, *p* = 0.03). These results suggest that aerobic fitness, expressed as a treadmill time or as 60s HRR, is partially mediated by participation in MVPA and that these models explained 32% (R^2^ = 0.32) and 45% (R^2^ = 0.45) of the variance in treadmill time and 60 s HRR, respectively. Lastly, for SBP, there was a direct effect of physical literacy on MVPA (β = 3.22, 95% CI: 1.98–4.46, *p* < 0.001), a non-significant direct effect of MVPA on SBP (β = 0.02, 95% CI: −0.02, 0.06, *p* = 0.31), and a non-significant indirect effect of MVPA on the relationship between physical literacy and SBP (β = 0.06, 95% CI: −0.06, 0.18, *p* = 0.32). This model explained 33% of the variance (R^2^ = 0.33) in SBP. 

## 4. Discussion

This is one of the first studies to empirically assess the relationships between physical literacy and health (body composition, fitness, blood pressure, and HRQOL) in school-aged children. The physical literacy composite score, a combination of PLAYfun, PLAYself, and PLAYparent, was associated with all health indicators. The strongest association was observed between PL and 60s HRR, an indicator of aerobic fitness. It was also determined that MVPA mediated the associations between physical literacy and aerobic fitness, as indicated by either treadmill time or 60 s HRR. These findings provide initial support for theories that position physical literacy as a determinant of health across the lifespan [6]. Evidence to support the proposed associations between physical literacy and health is necessary to move this field beyond physical education, recreation, and sport. 

To date, most research on the topic of physical literacy and health in children was conducted with a cross-Canadian sample of over 10,000 school-age children using the CAPL to assess physical literacy and field-based measures of health indicators [11,36]. A weak relationship was observed between indicators of aerobic fitness and children’s perceived adequacy and predilection toward physical activity, but not with other components of the CAPL [37]. The physical competence domain of CAPL and the total CAPL scores were associated with cardiorespiratory fitness, assessed with the PACER 20 m shuttle run test [14], similar to the associations we observed between physical literacy composite score and aerobic fitness. Lastly, it was observed that children who were a healthy weight had higher CAPL scores than children who were overweight or obese [12]. In the current study, while we did not classify participants based on weight status, we did observe that physical literacy was negatively associated with %BF. 

To represent physical literacy, a composite score of PLAYfun, PLAYparent, and PLAYself was generated. In a review of 50 studies, core attributes of physical literacy were identified as movement competence, motivation, confidence, self-esteem, knowledge and understanding, and value and responsibility for physical activity [8]. This work, in addition to the International Consensus Statement on Physical Literacy, suggests that a composite score, rather than a single PL assessment tool, may better reflect the multiple domains of PL. Through confirmatory factor analysis, it was determined that several domains (perceived competence, motivation, enjoyment, and motor skills), work synergistically to produce physical literacy in school-age children; however, the domains were assessed with tools not specially designed to measure components of physical literacy [38]. In previous work, a composite physical literacy score was generated from a combination of physical literacy measures (PLAYfun) and validated questionnaires that assessed motivation, confidence, knowledge, and understanding related to physical activity. In that study, the physical literacy composite score increased in the intervention group and decreased in the control group following 11 weeks of physical literacy-enriched programming for university students [39]. Our study is the first study to generate a physical literacy composite score using the PLAY Tools, that were specifically designed to assess physical literacy. 

Physical literacy is proposed to be the foundation of an active future and as a precursor to physical activity participation [2]. It has been observed that children who met the Canadian physical activity guideline of 60 min of daily MVPA displayed a higher physical competence and motivation and confidence physical literacy domain scores, as measured by the CAPL [13]. When physical literacy was assessed with PLAYfun, a positive relationship with pedometer-measured PA was reported (R^2^ = 0.30, *p* < 0.05) [16]. Similar relationships have been reported between PLAYfun and self-reported PA in another Canadian study [40]. The current findings confirm that MVPA, measured objectively with accelerometers, is associated with the physical literacy composite score. While it is beneficial to better understand the associations between physical literacy and physical activity, it remains to be determined how physical literacy can be effectively fostered in children and youths. The next significant step in this field is to design and implement interventions based on the concepts of physical literacy and assess if they have an impact on physical activity levels and contribute to better health indicators. The results of this study suggest that physical literacy is associated with indicators of aerobic fitness, and that this relationship is influenced by children’s participation in MVPA. 

This study has potential limitations that should be addressed in future research. This study was conducted in Canadian children, and results may not be comparable to other populations. The composite score of PL was novel and psychometric properties are not available; therefore, it is unclear if the method used to combine PLAYfun, PLAYparent, and PLAYself was most appropriate. Based on the definition of physical literacy [41], it was not appropriate to use the scores of PLAYfun, PLAYparent, and PLAYself individually. Future work should consider how various PLAY tools can be combined into one score that reflects the multiple domains of physical literacy. With this study’s cross-sectional design, it was not possible to determine the causal relationships between physical literacy, physical activity, and health. Mediation analysis would have been more appropriate if the physical literacy, physical activity, and health indicators were not measured at the same timepoint, but the results do help us understand these novel relationships that have not previously been investigated. Aerobic fitness was not assessed with the gold standard, VO2max, because the methodology of this study was developed for the participant’s young age (3 to 5-years-old) at the beginning of the study [17]. The direct measurement of VO2max would not have been feasible in that young sample. Rather, time to exhaustion with the modified Bruce Protocol was used, and is strongly correlated with direct VO2max in children [27]. The accelerometers were not waterproof, and participants were asked to remove the devices for swimming, therefore underestimating the physical activity levels of some participants. Lastly, success on the PLAYfun or aerobic fitness assessments could have been impacted by a participant’s motivation, and not all participants were similarly motivated to perform their best on the assessment, despite the continued encouragement from the assessors.

## 5. Conclusions

The present study determined that physical literacy was associated with health, represented as body composition, fitness, blood pressure, and HRQOL, and that the association between PL and aerobic fitness was mediated by MVPA. To our knowledge, this was the first study to explore these relationships using the PLAY Tools to assess physical literacy and lab-based measures to assess health. This study generated novel information that can inform future research in this field and contribute to the growing evidence that PL is the foundation for an active future [2], and that physical literacy is correlated with several health indicators [6]. Future research is needed to explore these relationships over time, and in additional demographics. 

## Figures and Tables

**Table 1 ijerph-17-05367-t001:** Participant demographics and descriptive statistics.

	Whole Sample			Girls	Boys	*p*-Value
	Mean (SD)	Min	Max	Mean (SD)	Mean (SD)	
Age (years)	10.76 (1.04)	8.38	13.66	10.83 (1.01)	10.69 (1.07)	0.330
YPHV (years)	−1.75 (1.24)	−4.07	1.61	−0.96 (1.01)	−2.57 (0.86)	<0.001 *
Height (cm)	145.71 (9.36)	124.45	174.65	145.78 (9.92)	145.63 (8.79)	0.904
Weight (kg)	38.03 (10.38)	22.30	91.75	38.53 (10.43)	37.52 (10.35)	0.470
BMI (m/kg^2^)	17.68 (3.13)	13.02	31.40	17.89 (3.03)	17.45 (3.23)	0.294
BMI%ile	47.87 (30.55)	0.44	99.04	49.52 (30.17)	46.17 (31.0)	0.415
Percent Body Fat	21.08 (6.60)	3.10	39.71	22.93 (6.18)	19.16 (6.50)	<0.001 *
Treadmill Time (min)	10.91 (2.88)	4.27	21.47	10.36 (2.68)	11.47 (2.98)	0.005 *
60 s HRR (bpm)	56 (14)	27	100	50 (13)	61 (12)	<0.001 *
SBP (mmHg)	100 (7)	84	125	100 (7)	101.0 (7)	0.103
HRQOL	76.83 (11.27)	30.43	96.74	78.68 (11.42)	74.93 (10.84)	0.014 *
MVPA (min/day)	62.39 (21.07)	18.43	124.86	56.73 (19.91)	68.39 (20.70)	<0.001 *

SD: standard deviation; YPHV: years from peak height velocity; BMI: body mass index; HR: heart rate; HRR: heart rate recovery; bpm: beats per minute; HRQOL: health-related quality of life. *p*-value represents the results of independent *t*-tests to determine differences between boys and girls; * *p* < 0.05.

**Table 2 ijerph-17-05367-t002:** PLAYfun, PLAYself, PLAYparent, and physical literacy composite scores.

	Whole Sample			Girls	Boys	*p*-Value
	Mean (SD)	Min	Max	Mean (SD)	Mean (SD)	
PLAYfun	49.09 (7.56)	21.22	67.67	47.91 (7.13)	50.33 (7.82)	0.017 *
PLAYself	63.50 (10.54)	39.43	97.52	73.66 (10.76)	73.33 (10.35)	0.820
PLAYparent	127.98 (16.0)	76.27	149.91	127.13 (15.94)	128.87 (16.08)	0.423
Physical Literacy Composite Score	−0.007 (2.19)	−8.78	5.56	−0.22 (2.05)	0.22 (2.33)	0.151

SD: standard deviation; PLAY: Physical Literacy Assessment for Youth. *p*-value represents the results of independent *t*-tests to determine differences between boys and girls; * *p* < 0.05.

**Table 3 ijerph-17-05367-t003:** Associations between physical literacy composite score and percent body fat, physical activity, aerobic fitness, blood pressure, and HRQOL.

	β (95% CI)	T Statistic	*p*-Value	R^2^
Percent Body Fat				0.228
Physical literacy composite	−0.56 (−0.93, −1.94)	−3.02	0.003	
Sex	−0.18 (2.36, 2.00)	−0.16	0.869	
YPHV	2.27 (1.340, 3.14)	5.13	<0.001	
Constant	25.16 (23.76, 26.55)	35.59	<0.001	
MVPA				0.235
Physical literacy composite	3.19 (2.00, 4.40)	5.25	<0.001	
Sex	6.66 (−0.58, 13.90)	1.81	0.071	
YPHV	−3.46 (−6.35, −0.56)	−2.35	0.020	
Constant	53.91 (49.36, 58.45)	23.39	<0.001	
Treadmill Time				0.212
Physical literacy composite	0.52 (0.36, 0.69)	6.21	<0.001	
Sex	0.61 (−0.36, 1.57)	0.49	0.219	
YPHV	−0.33 (−0.72, 0.07)	0.20	0.101	
Constant	10.01 (9.38, 10.65)	0.32	<0.001	
60 s HRR				0.357
Physical literacy composite	0.92 (0.22, 1.61)	2.59	0.010	
Sex	0.44 (−3.62, 4.51)	0.22	0.829	
YPHV	−6.12 (−7.76, −4.48)	−7.37	<0.001	
Constant	44.38 (41.71, 47.04)	32.80	<0.001	
Systolic Blood Pressure				0.109
Physical literacy composite	−0.54 (−0.93, −0.15)	−2.73	0.007	
Sex	4.40 (2.08, 6.72)	3.74	<0.001	
YPHV	2.04 (1.11, 2.97)	0.472	<0.001	
Constant	101.56 (100.08, 103.05)	134.93	<0.001	
HRQOL				0.156
Physical literacy composite	1.73 (1.05, 2.40)	5.06	<0.001	
Sex	−3.68 (−7.76, 0.40)	−1.78	0.077	
YPHV	0.68 (−0.96, 2.33)	0.83	0.412	
Constant	79.70 (77.13, 82.26)	61.32	<0.001	

YPHV: years from peak height velocity, MVPA: moderate-to-vigorous physical activity; HRQOL: health-related quality of life.

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
