# Peer review of "Physical Literacy, Physical Activity, and Health Indicators in School-Age Children"

_ijerph, 2020, doi:10.3390/ijerph17155367_

Round 1

Reviewer 1 Report

General comments:

This study examined relationships between physical literacy and various indicators of health in 9-11-year-old children. The main findings were that improved physical literacy (using a composite approach) was associated with several favourable health outcomes, and physical literacy and aerobic fitness were associated with moderate-to-vigorous physical activity.

This is a well-written study that utilised several methods to explore the topic of interest. The authors used a relatively large sample size for this type of study for which they should be commended. The findings lay the foundations for other studies to build on their data (and maybe collaborative data from other countries as well). I only have some minor comments.

Specific comments

  • Line 32: the four elements should either be in brackets or using “-“ and “-“
  • Line 51-55: add reference(s) for this statement
  • Line 81 - specify the age range of schoolchildren (e.g. 9 to 11 years)
  • Line 137 - were adequate measures taken to ensure children had not eaten or drunk any liquid prior to measurement?
  • Stats analysis - add p-value and whether data are expressed as mean +/- SD (or median and range)
  • Line 239 onwards - please add p-values for these significant differences (in brackets)
  • Table 1 - if the data was not normally distributed then median and range should be shown
  • Results - no need to keep stating 'significantly' if actual p-value (i.e. p<0.05) is shown. And, some values show e.g. F-stat whereas others do not; what's the rationale for this?
  • Line 267 - mediation (spelling)
  • Line345-361 - this data has been conducted in Canadian children; should be comparable for some (e.g. other western countries) but maybe not all

Author Response

Dear Reviewer #1,

Thank you for your kind and thoughtful remarks regarding our study of physical literacy, physical activity and health indicators in school-age children. Please see our responses to your comments below.

Line 32: the four elements should either be in brackets of using “-“ and “-“.

Brackets are now added to this sentence on lines 31-33; “This definition of physical literacy includes four interconnected elements (affective, physical, cognitive and behavioural) that change and adapt across the lifespan [3]”.

Line 51-55: add reference(s) for this statement.

Thank you for noticing this omission, the appropriate reference is now added.

Line 81 – specify the age range of schoolchildren (e.g. 9 to 11 years)

The age range of participants (8-13-year-old children) is now added in the purpose statement (line 83).

Line 137 – were adequate measures taken to ensure children had not eater or drunk any liquid prior to measurement?

Thank you for your comment. Children were not fasted prior to measurements, with the exception of blood pressure. Participants completed an assessment of cardiovascular health immediately prior to their visit to the exercise laboratory. They were fasted for 3 hours (water was allowed) prior to the cardiovascular health visit and were then welcome to eat a light snack prior to their visit to the exercise laboratory. 

Stats analysis – add p-value and whether data are expressed as mean +/- SD (or median and range).

Thank you for this comment, these are now added: “A p-value of 0.05 was used to specific statistical significance. Descriptive statistics (mean, standard deviation, minimum and maximum) of the participant’s age, YPHV, sex, percent body fat, treadmill time, 60-sec HRR, blood pressure, HRQOL, and PLAY Tools were calculated” (lines 190-193).

Line 239 onwards- please add p-values for these significant differences (in brackets).

Thank you for this comment, the p-values are now added in brackets in the text of the results section.

Table 1 – if the data was not normally distributed then median and range should be shown.

The minimum and maximum values are included in Table 1. The median values for the non-normally distributed variables are now included in the text of the results.

Results- no need to keep stating 'significantly' if actual p-value (i.e. p<0.05) is shown. And, some values show e.g. F-stat whereas others do not; what's the rationale for this?

After adding the p-values, the term ‘significant’ was removed from the text of the results. The F-stat was included for the results of all linear regression analyses. Thank you for the comments to update the results section.

Line 267 - mediation (spelling)

Thank you for catching this spelling error, it is now corrected in the text.

Lines 345-361 - this data has been conducted in Canadian children; should be comparable for some (e.g. other western countries) but maybe not all

Thank you for sharing this comment, it has no been addressed in the limitations section "This study was conducted in Canada children, and results may not be comparable to other populations."

Sincerely,

Dr. Brian Timmons & co-authors

Reviewer 2 Report

Manuscript is very well written, thorough and concise.  Topic is of great relevance to educators, health professionals and others interested in the relationship between PL, health and movement.

No concerns, however one question.  The development of the PL composite score is briefly described, with what appears to be equal weight to the three contributing sources, PLAYfun, PLAYparent & PLAYself.  Justification of why this approach was adopted should be included.  This is noted as a limitation of the study but clear justification of the approach would be valuable addition.

It is noted that participants were not coded based on BMI - under, average or overweight, for example - however separating participants into groups may provide further insight in the relationships between factors examined.  Just a thought, not something I feel strongly should be included here.

Author Response

Dear Reviewer #2,

Thank you for your kind and thoughtful remarks regarding our study of physical literacy, physical activity and health indicators in school-age children. Please see our responses to your comments below.

Manuscript is very well written, thorough and concise.  Topic is of great relevance to educators, health professionals and others interested in the relationship between PL, health and movement.

Thank you for the kind words, we are excited to share this manuscript with researchers, educators, health professionals and other practitioners.

No concerns, however one question.  The development of the PL composite score is briefly described, with what appears to be equal weight to the three contributing sources, PLAYfun, PLAYparent & PLAYself.  Justification of why this approach was adopted should be included.  This is noted as a limitation of the study but clear justification of the approach would be valuable addition.

Thank you for this comment. Yes, the PL composite score gives equal weight to PLAYfun, PLAYparent and PLAYself. Considering the definition of physical literacy, we determined it was not appropriate to use the PLAYfun, PLAYself or PLAYparent scores individually. At this point, the psychometric properties of a combined physical literacy score with the PLAY Tools has not been explored. The limitations section of the discussion has been updated to include further justification of this approach. “Based on the definition of physical literacy [41], it was not appropriate to use the scores of PLAYfun, PLAYparent and PLAYself individually. Future work should consider how various PLAY Tools can be combined into one score that reflects the multiple domains of physical literacy” is now included on lines 454-456.

It is noted that participants were not coded based on BMI - under, average or overweight, for example - however separating participants into groups may provide further insight in the relationships between factors examined.  Just a thought, not something I feel strongly should be included here.

Great comment and question. We considered this approach; however, our analyses would not have been appropriate for the small number of participants that would be in the underweight, overweight or obese groups. In this sample, we did not observe a significant relationship between physical literacy and BMI or BMI percentile. This would be very interesting to explore in future studies and we will consider this approach should our sample size be larger.

Sincerely,

Dr. Brian Timmons & co-authors